# U-Time: A Fully Convolutional Network for Time Series Segmentation Applied to Sleep Staging

**Mathias Perslev**
Department of Computer Science
University of Copenhagen
map@di.ku.dk

**Michael Hejselbak Jensen**
Department of Computer Science
University of Copenhagen
mhejselbak@gmail.com

**Sune Darkner**
Department of Computer Science
University of Copenhagen
darkner@di.ku.dk

**Poul Jørgen Jennum**
Danish Center for Sleep Medicine
Rigshospitalet, Denmark
poul.joergen.jennum@regionh.dk

**Christian Igel**
Department of Computer Science
University of Copenhagen
igel@diku.dk

## Abstract

Neural networks are becoming more and more popular for the analysis of physiological time-series. The most successful deep learning systems in this domain combine convolutional and recurrent layers to extract useful features to model temporal relations. Unfortunately, these recurrent models are difficult to tune and optimize. In our experience, they often require task-specific modifications, which makes them challenging to use for non-experts. We propose U-Time, a fully feed-forward deep learning approach to physiological time series segmentation developed for the analysis of sleep data. U-Time is a temporal fully convolutional network based on the U-Net architecture that was originally proposed for image segmentation. U-Time maps sequential inputs of arbitrary length to sequences of class labels on a freely chosen temporal scale. This is done by implicitly classifying every individual time-point of the input signal and aggregating these classifications over fixed intervals to form the final predictions. We evaluated U-Time for sleep stage classification on a large collection of sleep electroencephalography (EEG) datasets. In all cases, we found that U-Time reaches or outperforms current state-of-the-art deep learning models while being much more robust in the training process and without requiring architecture or hyperparameter adaptation across tasks.

## 1 Introduction

During sleep our brain goes through a series of changes between different *sleep stages*, which are characterized by specific brain and body activity patterns [Kales and Rechtschaffen, 1968, Iber and AASM, 2007]. *Sleep staging* refers to the process of mapping these transitions over a night of sleep. This is of fundamental importance in sleep medicine, because the sleep patterns combined with other variables provide the basis for diagnosing many sleep related disorders [Sateia, 2014]. The stages can be determined by measuring the neuronal activity in the cerebral cortex (via electroencephalography, EEG), eye movements (via electrooculography, EOG), and/or the activity of facial muscles (via electromyography, EMG) in a *polysomnography* (PSG) study (see Figure S.1 in the Supplementary

Material). The classification into stages is done manually. This is a difficult and time-consuming process, in which expert clinicians inspect and segment the typically 8–24 hours long multi-channel signals. Contiguous, fixed-length intervals of 30 seconds are considered, and each of these *segments* is classified individually.

Algorithmic sleep staging aims at automating this process. Recent work shows that such systems can be highly robust (even compared to human performance) and may play an important role in developing novel biomarkers for sleep disorders and other (e.g., neurodegenerative and psychiatric) diseases [Stephansen et al., 2018, Warby et al., 2014, Schenck et al., 2014]. Deep learning is becoming increasingly popular for the analysis of physiological time-series [Faust et al., 2018] and has already been applied to sleep staging [Robert et al., 1998, Ronzhina et al., 2012, Faust et al., 2019]. Today's best systems are based on a combination of convolutional and recurrent layers [Supratak et al., 2017, Biswal et al., 2017]. While recurrent neural networks are conceptually appealing for time series analysis, they are often difficult to tune and optimize in practice, and it has been found that for many tasks across domains recurrent models can be replaced by feed-forward systems without sacrificing accuracy [Bai et al., 2018, Chen and Wu, 2017, Vaswani et al., 2017].

This study introduces U-Time, a feed-forward neural network for sleep staging. U-Time as opposed to recurrent architectures can be directly applied across datasets of significant variability without any architecture or hyperparameter tuning. The task of segmenting the time series is treated similar to image segmentation by the popular U-Net architecture [Ronneberger et al., 2015]. This allows segmentation of an entire PSG in a single forward pass and to output sleep stages at any temporal resolution. Fixing a temporal embedding, which is a common argument against feed-forward approaches to time series analysis, is no problem, because in our setting the full time series is available at once and is processed entirely (or in large chunks) at different scales by the special network architecture.

In the following, we present our general approach to classifying fixed length continuous segments of physiological time series. In Section 3, we apply it to sleep stage classification and evaluate it on 7 different PSG datasets using a fixed architecture and hyperparameter set. In addition, we performed many experiments with a state-of-the-art recurrent architecture, trying to improve its performance over U-Time and to assess its robustness against architecture and hyperparameter changes. These experiments are listed in the Supplementary Material. Section 4 summarizes our main findings, before we conclude in Section 5.

## 2 Method

U-Time is a fully convolutional encoder-decoder network. It is inspired by the popular U-Net architecture originally proposed for image segmentation [Ronneberger et al., 2015, Koch et al., 2019b, Perslev et al., 2019] and so-called temporal convolutional networks [Lea et al., 2016]. U-Time adopts basic concepts from U-Net for 1D time-series segmentation by mapping a whole sequence to a dense segmentation in a single forward pass.

Let $\mathbf{x} \in \mathbb{R}^{\tau S \times C}$ be a physiological signal with $C$ channels sampled at rate $S$ for $\tau$ seconds. Let $e$ be the frequency at which we want to segment $\mathbf{x}$, that is, the goal is to map $\mathbf{x}$ to $\lfloor \tau \cdot e \rfloor$ labels, where each label is based on $i = S/e$ sampled points. In sleep staging, 30 second intervals are typically considered (i.e., $e = 1/30$ Hz). The input $x$ to U-Time are $T$ fixed-length connected *segments* of the signal, each of length $i$. U-Time predicts the $T$ labels at once. Specifically, the model $f(x; \theta) : \mathbb{R}^{T \times i \times C} \to \mathbb{R}^{T \times K}$ with parameters $\theta$ maps $x$ to class confidence scores for predicting $K$ classes for all $T$ segments. That is, the model processes 1D signals of length $t = Ti$ in each channel.

The segmentation frequency $e$ is variable. For instance, a U-Time model trained to segment with $e = 1/30$ Hz may output sleep stages at a higher frequency at inference time. In fact, the extreme case of $e = S$, in which every individual time-point of $x$ gets assigned a stage, is technically possible, although difficult (or even infeasible) to evaluate (see for example Figure 3). U-Time, in contrast to other approaches, allows for this flexibility, because it learns an intermediate representation of the input signal where a confidence score for each of the $K$ classes is assigned to each time point. From this dense segmentation the final predictions over longer segments of time are computed by projecting the fine-grained scores down to match the rate $e$ at which human annotated labels are available.

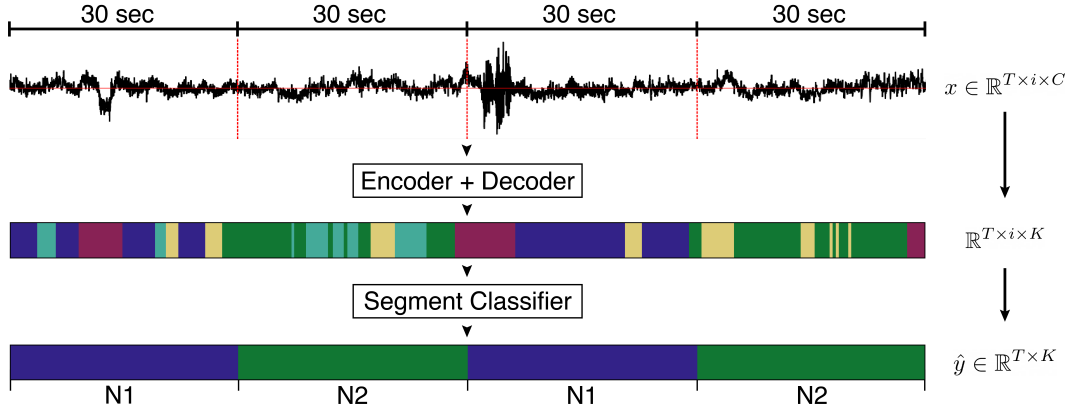

Figure 1: Illustrative example of how U-Time maps a potentially very long input sequence (here only $T = 4$ for visual purposes) to segmentations at a chosen temporal scale (here $e = 1/30$ Hz) by first segmenting the signal at every data-point and then aggregating these scores to form final predictions.

The U-Time model $f$ consists of three logical submodules: The encoder $f_{\text{enc}}$ takes the raw physiological signal and represents it by a deep stack of feature maps, where the input is sub-sampled several times. The decoder $f_{\text{dec}}$ learns a mapping from the feature stack back to the input signal domain that gives a dense, point-wise segmentation. A segment classifier $f_{\text{segment}}$ uses the dense segmentation to predict the final sleep stages at a chosen temporal resolution. These steps are illustrated in Figure 1. An architecture overview is provided in Figure 2 and detailed in Supplementary Table S.2.

**Encoder** The encoder consists of four convolution blocks. All convolutions in the three submodules preserve the input dimensionality through zero-padding. Each block in the encoder performs two consecutive convolutions with 5-dimensional kernels dilated to width 9 [Yu and Koltun, 2015] followed by batch normalization [Ioffe and Szegedy, 2015] and max-pooling. In the four blocks, the pooling windows are 10, 8, 6, and 4, respectively. Two additional convolutions are applied to the fully down-sampled signal. The aggressive down-sampling reduces the input dimensionality by a factor 1920 at the lowest layers. This 1) drastically reduces computational and memory requirements even for very long inputs, 2) enforces learning abstract features in the bottom layers and, 3), combined with stacked dilated convolutions, provides a large receptive field at the last convolution layer of the encoder. Specifically, the maximum theoretical receptive field of U-Time corresponds to approx. 5.5 minutes given a 100 Hz signal (see Luo et al. [2017] for further information on theoretical and effective receptive fields).

The input $x$ to the encoder could be an entire PSG record ($T = \lfloor \tau \cdot e \rfloor$) or a subset. As the model is based on convolution operations, the total input length $t$ need not be static either, but could change between training and testing or even between individual mini-batches. While $t$ is adjustable, it must be large enough so that all max-pooling operations of the encoder are defined, which in our implementation amounts to $t_{\min} = 1920$ or 19.2 seconds of a 100 Hz signal. A too small $t$ reduces performance by preventing the model from exploiting long-range temporal relations.

**Decoder** The decoder consists of four transposed-convolution blocks [Long et al., 2014], each performing nearest-neighbour up-sampling [Odena et al., 2016] of its input followed by convolution with kernel sizes 4, 6, 8 and 10, respectively, and batch normalization. The resulting feature maps are concatenated (along the filter dimension) with the corresponding feature maps computed by the encoder at the same scale. Two convolutional layers, both followed by batch normalization, process the concatenated feature maps in each block. Finally, a point-wise convolution with $K$ filters (of size 1) results in $K$ scores for each sample of the input sequence.

In combination, the encoder and decoder maps a $t \times C$ input signal to $t \times K$ confidence scores. We may interpret the decoder output as class confidence scores assigned to every sample point of the input signal, but in most applications we are not able to train the encoder-decoder network in a supervised setting as labels are only provided or even defined over segments of the input signal.

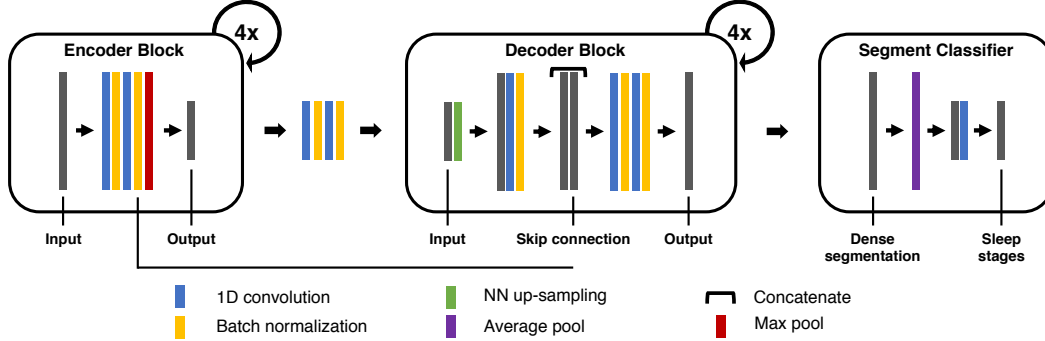

Figure 2: Structural overview of the U-Time architecture. Please refer to Supplementary Figure S.2 for an extended, larger version.

**Segment classifier**    The segment classifier serves as a trainable link between the intermediate representation defined by the encoder-decoder network and the label space. It aggregates the sample-wise scores to predictions over longer periods of time. For periods of $i$ time steps, the segment classifier performs channel-wise mean pooling with width $i$ and stride $i$ followed by point-wise convolution (kernel size 1). This aggregates and re-weights class confidence scores to produce scores of lower temporal resolution. In training, where we only have $T$ labels available, the segment classifier maps the dense $t \times K$ segmentation to a $T \times K$-dimensional output.

Because the segment classifier relies on the mean activation over a segment of decoder output, learning the full function $f$ (encoder+decoder+segment classifier) drives the encoder-decoder sub-network to output class confidence scores distributed over the segment. As the input to the segment classifier does not change in expectation if $e$ (the segmentation frequency) is changed, this allows to output classifications on shorter temporal scales at inference time. Such scores may provide important insight into the individual sleep stage classifications by highlighting regions of uncertainty or fast transitions between stages on shorter than 30 second scales. Figure 3 shows an example.

## 3    Experiments and Evaluation

Our brain is in either an awake or sleeping state, where the latter is further divided into rapid-eye-movement sleep (REM) and non-REM sleep. Non-REM sleep is further divided into multiple states. In his pioneering work, Kales and Rechtschaffen [1968] originally described four non-REM stages, S1, S2, S3 and S4. However, the American Academy of Sleep Medicine (AASM) provides a newer characterization [Iber and AASM, 2007], which most importantly changes the non-REM naming convention to N1, N2, and N3, grouping the original stages S3 and S4 into a single stage N3. We use this 5-class system and refer to Table S.1 in the Supplementary Material for an overview of primary features describing each of the AASM sleep stages.

We evaluated U-Time for sleep-stage segmentation of raw EEG data. Specifically, U-Time was trained to output a segmentation of an EEG signal into $K = 5$ sleep stages according to the AASM, where each segment lasts 30 seconds ($e = 1/30$ Hz). We fixed $T = 35$ in our experiments. That is, for a $S = 100$ Hz signal we got an input of $t = 105000$ samples spanning 17.5 minutes.

Our experiments were designed to gauge the performance of U-Time across several, significantly different sleep study cohorts when no task-specific modifications are made to the architecture or hyperparameters between each. In the following, we describe the data pre-processing, optimization, and evaluation in detail, followed by a description of the datasets considered in our experiments.

**Preprocessing**    All EEG signals were re-sampled at $S = 100$ Hz using polyphase filtering with automatically derived FIR filters. Across the datasets, sleep stages were scored by at least one human expert at temporal resolution $e = 1/30$ Hz. When stages were scored according to the Kales and Rechtschaffen [1968] manual, we merged sleep stages S3 and S4 into a single N3 stage to comply with the AASM standard. We discarded the rare and typically boundary-located sleep stages such as 'movement' and 'non-scored' and their corresponding PSG signals, producing the identical label set

{W, N1, N2, N3, R} for all the datasets. EEG signals were individually scaled for each record to median 0 and inter quartile range (IQR) 1.

Some records display extreme values typically near the start or end of the PSG studies when electrodes are placed or the subject is entering or leaving the bed. To stabilize the pre-processing scaling as well as learned batch normalization, all 30 second segments that included one or more values higher than 20 times the global IQR of that record were set to zero. Note that this only applied if the segment was scored by the human observer (almost always classified 'wake' as these typically occur outside the 'in-bed' region), as they would otherwise be discarded. We set the values to zero instead of discarding them to maintain temporal consistency between neighboring segments.

**Optimization**   U-Time was optimized using a fixed set of hyperparameters for all datasets. We used the Adam optimizer [Kingma and Ba, 2014] with learning rate $\eta = 5 \cdot 10^{-6}$ minimizing the generalized dice cost function with uniform class weights [Sudre et al., 2017, Crum et al., 2006], $\mathcal{L}(y, \hat{y}) = 1 - \frac{2}{K} \frac{\sum_k^K \sum_n^N y_{kn} \hat{y}_{kn}}{\sum_k^K \sum_n^N y_{kn} + \hat{y}_{kn}}$. This cost function is useful in sleep staging, because the classes may be highly imbalanced. To further counter class imbalance we selected batches of size $B = 12$ on-the-fly during training according to the following scheme: 1) we uniformly sample a class from the label set {W, N1, N2, N3, R}, 2) we select a random sleep period corresponding to the chosen class from a random PSG record in the dataset, 3) we shift the chosen sleep segment to a random position within the $T = 35$ width window of sleep segments. This scheme does not fully balance the batches, as the 34 remaining segments of the input window are still subject to class imbalance.

Training of U-Time was stopped after 150 consecutive epochs of no validation loss improvement (see also Cross-validation below). We defined one epoch as $\lceil L/T/B \rceil$ gradient steps, where $L$ is the total number of sleep segments in the dataset, $T$ is the number of fixed-length connected segments input to the model and $B$ is the batch size. Note that we found applying regularization unnecessary when optimizing U-Time as overfitting was negligible even on the smallest of datasets considered here (see Sleep Staging Datasets 3 below).

**Model specification and hyperparameter selection**   The encoder and decoder parts of the U-Time architecture are 1D variants of the 2D U-Net type model that we have found to perform excellent across medical image segmentation problems (described in [Koch et al., 2019b, Perslev et al., 2019]). However, U-Time uses larger max-pooling windows and dilated convolution kernels. These changes were introduced in order to increase the theoretical receptive field of U-Time and were made based on our physiological understand of sleep staging rather than hyperparameter tuning. The only choice we made based on data was the loss function, where we compared dice loss and cross entropy using 5-fold cross-validation on the Sleep-EDF-39 dataset (see below). We did not modify the architecture or any hyperparameters (e.g., learning rates) after observing results on any of the remaining datasets. Our minimal hyperparameter search minimizes the risk of unintentional method-level overfitting.

U-Time as applied here has a total of $\approx 1.2$ million trainable parameters. Note that this is at least one order of magnitude lower than typical CNN-LSTM architectures such as DeepSleepNet [Supratak et al., 2017]. We refer to Table S.2 and Figure S.2 in the Supplementary Material for a detailed model specification as well as to Table S.3 in the Supplementary Material for a detailed list of hyperparameters.

**Cross-validation**   We evaluated U-Time on 7 sleep EEG datasets (see below) with no task-specific architectural modifications. For a fair comparison with published results, we adopted the evaluation setting that was most frequent in the literature for each dataset. In particular, we adopted the number of cross-validation (CV) splits, which are given in the results Table 2 below. All reported CV scores result from single, non-repeated CV experiments.

It is important to stress that CV was always performed on a per-subject basis. The entire EEG record (or multiple records, if one subject was recorded multiple times) were considered a single entity in the CV split process.[1] On all datasets except SVUH-UCD, $\lceil 5\% \rceil$ of the training records of each split were used for validation to implement early-stopping based on the validation F1 score [Sørensen,

1948, Dice, 1945]. For SVUH-UCD, a fixed number of training epochs (800) was used in all splits, because the dataset is too small to provide a representative validation set.

**Evaluation & metrics**   In Table 2 we report the per-class F1/dice scores computed over raw confusion matrices summed across all records and splits. This procedure was chosen to be comparable to the relevant literature. The table summarizes our results and published results for which the evaluation strategy was described clearly. Specifically, we only compare to studies in which CV has been performed on a subject-level and not segment level. In addition, we only compare to studies that either report F1 scores directly or provide other metrics or confusion matrices from which we could derive the F1 score. We only compare to EEG based methods.

**LSTM comparison**   We re-implemented the successful DeepSleepNet CNN-LSTM model [Supratak et al., 2017] for two purposes. First, we tried to push the performance of this model to the level of U-Time on the Sleep-EDF-39 and DCSM datasets (see below) through a series of hyperparameter experiments summarized in Table S.13 & Table S.14 in the Supplementary Material. Second, we used DeepSleepNet to establish a unified, state-of-the-art baseline. Because the DeepSleepNet system as introduced in Supratak et al. [2017] was trained for a fixed number of epochs without early stopping, we argue that direct application of the original implementation to new data would favour our U-Time model. Therefore, we re-implemented DeepSleepNet and plugged it into our U-Time training pipeline. This ensures that the models use the same early stopping mechanisms, class-balancing sampling schemes, and TensorFlow implementations. We employed pre- and finetune training of the CNN and CNN-LSTM subnetworks, respectively, as in Supratak et al. [2017]. We observed overfitting using the original settings, which we mitigated by reducing the default pre-training learning rate by a factor 10. For Sleep-EDF-39 and DCSM, DeepSleepNet was manually tuned in an attempt to reach maximum performance (see Supplementary Material). We did not evaluate DeepSleepNet on SVUH-UCD because of the small dataset size.

**Implementation**   U-Time is publicly available at `https://github.com/perslev/U-Time`. The software includes a command-line-interface for initializing, training and evaluating models through CV experiments automatically distributed and controlled over multiple GPUs. The code is based on TensorFlow [Abadi et al., 2015]. We ran all experiments on a NVIDIA DGX-1 GPU cluster using 1 GPU for each CV split experiment. However, U-Time can be trained on a conventional 8-12 GB memory GPU. Because U-Time can score a full PSG in a single forward-pass, segmenting 10+ hours of signal takes only seconds on a laptop CPU.

**Sleep Staging Datasets**   We evaluated U-Time on several public and non-public datasets covering many real-life sleep-staging scenarios. The PSG records considered in our experiments have been collected over multiple decades at multiple sites using various instruments and recording protocols to study sleep in both healthy and diseased individuals. We briefly describe each dataset and refer to the original papers for details. Please refer to Table 1 for an overview and a list of used EEG channels.

*Sleep-EDF*   A public PhysioNet database [Kemp et al., 2000, Goldberger et al., 2000] often used for benchmarking automatic sleep stage classification algorithms. As of 2019, the sleep-cassette subset of the database consists of 153 whole-night polysomnographic sleep recordings of healthy Caucasians age 25-101 taking no sleep-related medication. We utilze both the full Sleep-EDF database (referred to as *Sleep-EDF-153*) as well as a subset of 39 samples (referred to as *Sleep-EDF-39*) that correspond to an earlier version of the Sleep-EDF database that has been extensively studied in the literature. Note that for these two datasets specifically, we only considered the PSGs starting from 30 minutes before to 30 minutes after the first and last non-wake sleep stage as determined by the ground truth labels in order to stay comparable with literature such as Supratak et al. [2017].

*Physionet 2018*   The objective of the 2018 Physionet challenge [Ghassemi et al., 2018, Goldberger et al., 2000] was to detect arousal during sleep from PSG data contributed by the Massachusetts General Hospital's Computational Clinical Neurophysiology Laboratory. Sleep stages were also provided for the training set. We evaluated U-Time on splits of the 994 subjects in the training set.

*DCSM*   A non-public database provided by Danish Center for Sleep Medicine (DCSM), Rigshospitalet, Glostrup, Denmark comprising 255 whole-night PSG recordings of patients visiting the center for diagnosis of non-specific sleep related disorders. Subjects vary in demographic characteristics, diagnostic background and sleep/non-sleep related medication usage.

Table 1: Datasets overview. The Scoring column reports the annotation protocol (R&K = Rechtschaffen and Kales, AASM = American Academy of Sleep Medicine), Sample Rate lists the original rate (in Hz), and Size gives the number of subjects included in our study after exclusions.

| Dataset | Size | Sample Rate | Channel | Scoring | Disorders |
|---|---|---|---|---|---|
| S-EDF-39 | 39 | 100 | Fpz-Cz | R&K | None |
| S-EDF-153 | 153 | 100 | Fpz-Cz | R&K | None |
| Physio-2018 | 994 | 200 | C3-A2 | AASM | Non-specific sleep disorders |
| DCSM | 255 | 256 | C3-A2 | AASM | Non-specific sleep disorders |
| ISRUC | 99 | 200 | C3-A2 | AASM | Non-specific sleep disorders |
| CAP | 101 | 100-512 | C4-A1/C3-A2 | R&K | 7 types of sleep disorders |
| SVUH-UCD | 25 | 128 | C3-A2 | R&K | Sleep apnea, primary snoring |

*ISRUC* Sub-group 1 of this public database [Khalighi et al., 2016] comprises all-night PSG recordings of 100 adult, sleep disordered individuals, some of which were under the effect of sleep medication. Recordings were independently scored by two human experts allowing performance comparison between the algorithmic solution and human expert raters. We excluded subject 40 due to a missing channel.

*CAP* A public database [Terzano et al., 2002] storing 108 PSG recordings of 16 healthy subjects and 92 pathological patients diagnosed with one of bruxism, insomnia, narcolepsy, nocturnal frontal lobe epilepsy, periodic leg movements, REM behavior disorder, or sleep-disordered breathing. We excluded subjects brux1, nfle6, nfle25, nfle27, nfle33, n12 and n16 due to missing C4-A1 and C3-A2 channels or due to inconsistent meta-data information.

*SVUH-UCD* The St. Vincent's University Hospital / University College Dublin Sleep Apnea Database [Goldberger et al., 2000] contains 25 full overnight PSG records of randomly selected individuals under diagnosis for either obstructive sleep apnea, central sleep apnea or primary snoring.

## 4 Results

We applied U-Time with fixed architecture and hyperparameters to 7 PSG datasets. Table 2 lists the class-wise F1 scores computed globally (i.e., on the summed confusion matrices over all records) for U-Time applied to a single EEG channel (see Table 1), our re-implemented DeepSleepNet (CNN-LSTM) baseline and alternative models from literature. Table S.12 in the Supplementary material further reports a small number of preliminary multi-channel U-Time experiments, which we discuss below. Table S.5 to Table S.11 in the Supplementary Material display raw confusion matrices corresponding to the scores of Table 2. In Table S.4 in the Supplementary Material, we report the mean, standard deviation, minimum and maximum per-class F1 scores computed across individual EEG records, which may be more relevant from a practical perspective.

Even without task-specific modifications, U-Time reached high performance scores for large and small datasets (such as Physionet-18 and Sleep-EDF-39), healthy and diseased populations (such as Sleep-EDF-153 and DCSM), and across different EEG channels, sample rates, accusation protocols and sites etc. On all datasets, U-Time performed, to our knowledge, at least as well as any automated method from the literature that allows for a fair comparison – even if the method was tailored towards the individual dataset. In all cases, U-Time performed similar or better than the CNN-LSTM baseline.

We attempted to push the performance of the CNN-LSTM architecture of our re-implemented DeepSleepNet [Supratak et al., 2017] to the performance of U-Time on both the Sleep-EDF-39 and DCSM datasets. These hyperparameter experiments are given in Table S.13 and Table S.14 in the Supplementary Material. However, across 13 different architectural changes to the DeepSleepNet model, we did not observe any improvement over the published baseline version on the Sleep-EDF-39 dataset, indicating that the model architecture is already highly optimized for the particular study cohort. We found that relatively modest changes to the DeepSleepNet architecture can lead to large changes in performance, especially for the N1 and REM sleep stages. On the DCSM dataset, a smaller version of the DeepSleepNet (smaller CNN filters, specifically) improved performance slightly over the DeepSleepNet baseline.

Table 2: U-Time results across 7 datasets. U-Time and our CNN-LSTM baseline process single-channel EEG data. Referenced models process single- or multi-channel EEG data. References: [1] [Supratak et al., 2017], [2] [Vilamala et al., 2017], [3] [Phan et al., 2018], [4] [Tsinalis et al., 2016], [5] [Andreotti et al., 2018].

| | | Eval | | Global F1 scores | | | | | |
|---|---|---|---|---|---|---|---|---|---|
| Dataset | Model | Records | CV | W | N1 | N2 | N3 | REM | mean |
| S-EDF-39 | *U-Time* | 39 | 20 | 0.87 | 0.52 | 0.86 | 0.84 | 0.84 | 0.79 |
| | CNN-LSTM[1] | 39 | 20 | 0.85 | 0.47 | 0.86 | 0.85 | 0.82 | 0.77 |
| | VGGNet[2] | 39 | 20 | 0.81 | 0.47 | 0.85 | 0.83 | 0.82 | 0.76 |
| | CNN[3] | 39 | 20 | 0.77 | 0.41 | 0.87 | 0.86 | 0.82 | 0.75 |
| | Autoenc.[4] | 39 | 20 | 0.72 | 0.47 | 0.85 | 0.84 | 0.81 | 0.74 |
| S-EDF-153 | *U-Time* | 153 | 10 | 0.92 | 0.51 | 0.84 | 0.75 | 0.80 | 0.76 |
| | CNN-LSTM | 153 | 10 | 0.91 | 0.47 | 0.81 | 0.69 | 0.79 | 0.73 |
| Physio-18 | *U-Time* | 994 | 5 | 0.83 | 0.59 | 0.83 | 0.79 | 0.84 | 0.77 |
| | CNN-LSTM | 994 | 5 | 0.82 | 0.58 | 0.83 | 0.78 | 0.85 | 0.77 |
| DCSM | *U-Time* | 255 | 5 | 0.97 | 0.49 | 0.84 | 0.83 | 0.82 | 0.79 |
| | CNN-LSTM | 255 | 5 | 0.96 | 0.39 | 0.82 | 0.80 | 0.82 | 0.76 |
| ISRUC | *U-Time* | 99 | 10 | 0.87 | 0.55 | 0.79 | 0.87 | 0.78 | 0.77 |
| | CNN-LSTM | 99 | 10 | 0.84 | 0.46 | 0.70 | 0.83 | 0.72 | 0.71 |
| | Human obs. | 99 | - | 0.92 | 0.54 | 0.80 | 0.85 | 0.90 | 0.80 |
| CAP | *U-Time* | 101 | 5 | 0.78 | 0.29 | 0.76 | 0.80 | 0.76 | 0.68 |
| | CNN[5] | 104 | 5 | 0.77 | 0.35 | 0.76 | 0.78 | 0.76 | 0.68 |
| | CNN-LSTM | 101 | 5 | 0.77 | 0.28 | 0.69 | 0.77 | 0.75 | 0.65 |
| SVUH-UCD | *U-Time* | 25 | 25 | 0.75 | 0.51 | 0.79 | 0.86 | 0.73 | 0.73 |

# 5 Discussion and Conclusions

U-Time is a novel approach to time-series segmentation that leverages the power of fully convolutional encoder-decoder structures. It first implicitly segments the input sequence at every time point and then applies an aggregation function to produce the desired output.

We developed U-Time for sleep staging, and this study evaluated it on seven different sleep PSG datasets. For all tasks, we used the same U-Time network architecture and hyperparameter settings. This does not only rule out overfitting by parameter or structure tweaking, but also shows that U-Time is robust enough to be used by non-experts – which is of key importance for clinical practice. In all cases, the model reached or surpassed state-of-the-art models from the literature as well as our CNN-LSTM baseline. In our experience, CNN-LSTM models require careful optimization, which indicates that they may not generalize well to other cohorts. This is supported by the observed drop in CNN-LSTM baseline performance when transferred to, for example, the ISRUC dataset. We further found that the CNN-LSTM baseline shows large F1 score variations, in particular for sleep stage N1, for small changes of the architecture (see Table S.13 in the Supplementary Material). In contrast, U-Time reached state-of-the-art performance across the datasets without being tuned for each task. Our results show that U-Time can learn sleep staging based on various input channels across both healthy and diseased subjects. We attribute the general robustness of U-Time to its fully convolutional, feed-forward only architecture.

Readers not familiar with sleep staging should be aware that even human experts from the same clinical site may disagree when segmenting a PSG.[2] While human performance varies between datasets, the mean F1 overlap between typical expert annotators is at or slightly above 0.8 [Stephansen et al., 2018]. This is also the case on the ISRUC dataset as seen in Table 2. U-Time performs at the level of the human experts on the three non-REM sleep stages of the ISRUC dataset, while inferior

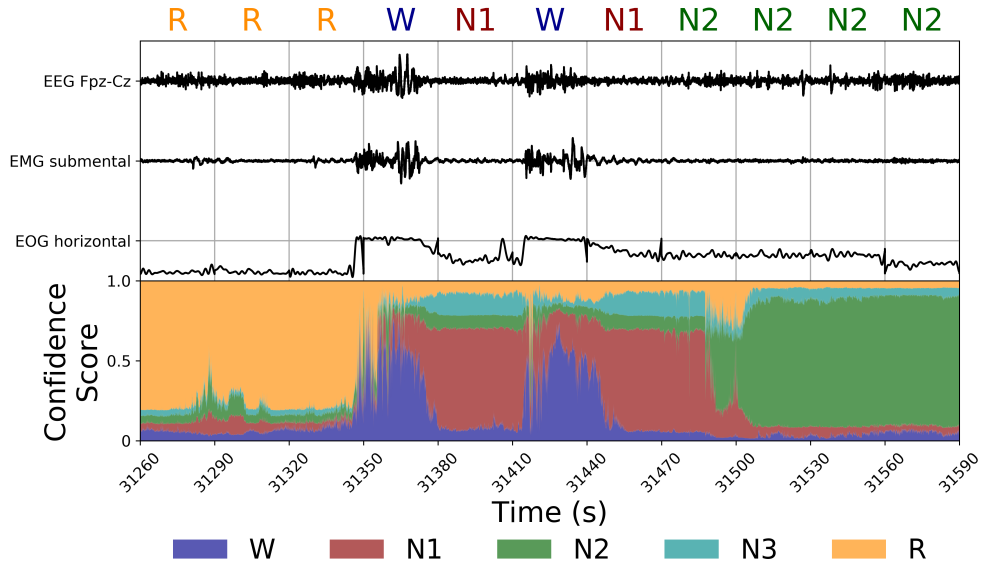

Figure 3: Visualization of the class confidence scores of U-Time trained on $C = 3$ input channels on the Sleep-EDF-153 dataset when the segmentation frequency $e$ is set to match the input signal frequency. Here, U-Time outputs 100 sleep stage scores per second. The top, colored letters give the ground truth labels for each 30 second segment. The height of the colored bars in the bottom frame gives the softmax (probability-like) scores for each sleep stage at each point in time.

on the REM sleep stage and slightly below on the wake stage. However, human annotators have the advantage of being able to inspect several channels including the EOG (eye movement), which often provides important information in separating wake and REM sleep stages. This is because the EEG activity in wake and REM stages is similar, while – as the name suggests – characteristic eye movements are indicative of REM sleep (see Table S.1 in the Supplementary Material). In this study we chose to use only a single EEG channel to compare to other single-channel studies in literature. It is highly likely that U-Time for sleep staging would benefit from receiving multiple input channels. This is supported by our preliminary multi-channel results reported in Supplementary Table S.12. On ISRUC and other datasets, the inclusion of an EOG channel improved classification of the REM sleep stage.

We observed the lowest U-Time performance on the CAP dataset, although on par with the model of Andreotti et al. [2018], which requires multiple input channels. The CAP dataset is difficult because it contains recordings from patients suffering from seven different sleep related disorders, each of which are represented by only few subjects, and because of the need for learning both the C4-A1 and C3-A2 channels simultaneously.

Besides its accuracy, robustness, and flexibility, U-Time has a couple of other advantageous properties. Being fully feed-forward, it is fast in practice as computations may be distributed efficiently on GPUs. The input window $T$ can be dynamically adjusted, making it possible to score an entire PSG record in a single forward pass and to obtain full-night sleep stage classifications almost instantaneously in clinical practice. Because of its special architecture, U-Time can output sleep stages at a higher temporal resolution than provided by the training labels. This may be of importance in a clinical setting for explaining the system's predictions as well as in sleep research, where sleep stage dynamics on shorter time scales are of great interest [Koch et al., 2019a]. Figure 3 shows an example.

While U-Time was developed for sleep staging, we expect its basic design to be readily applicable to other time series segmentation tasks as well. Based on our results, we conclude that fully convolutional, feed-forward architectures such as U-Time are a promising alternative to recurrent architectures for times series segmentation, reaching similar or higher performance scores while being much more robust with respect to the choice of hyperparameters.

## Footnotes

[1]Not doing so leads to data from the same subject being in both training and test sets and, accordingly, to overoptimistic results. This effect is very pronounced. Therefore, we do not discuss published results where training and test set were not split on a per-subject basis.

[2]This is true in particular for the N1 sleep stage, which is difficult to detect due to its transitional nature and non-strict separation from the awake and deep sleep stages.

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
