[Supplementary Material · 2469_camera_ready_supplementary.pdf]

## Supplementary Material

Table S.1: Brief characterization of typical features of the 5 sleep stages as defined by the AASM manual [Iber and AASM, 2007].

| Name | Encoding | Description |
|------|----------|-------------|
| Wake | W | Spans wakefulness to drowsiness. Consists of at least 50% alpha waves (8-13 Hz EEG signals). Rapid and reading eye movements. Eye blinks may occur. |
| Non-REM 1 | N1 | Short, light sleep stage comprising about 5%-10% of a night's sleep. Dominated by theta waves (4-7 Hz EEG signals). Slow eye movements in W $\rightarrow$ N1 transition. Some EMG activity, but lower than wake. |
| Non-REM 2 | N2 | Comprises 40%-50% of a normal night's sleep. EEG dispalys theta-waves like N1, but intercepted by so-called K-complexes and/or sleep spindles (short bursts of 13-16Hz EEG signal). |
| Non-REM 3 | N3 | Comprises about 20%-25% of a typical night's sleep. High amplitude, slow 0.3-3 Hz EEG signals. Low EMG activity. |
| REM | R | Rapid-eye-movements may occur. Displays both theta waves and alpha (like wake), but typically 1-2 Hz slower. EMG significantly reduced. Dreaming may occur this stage, which comprises 20%-25% of the night's sleep. |

Figure S.1: A segment of 30 seconds of a typical polysomnography (PSG) study showing multiple EOG, EEG, EMG and ECG channels. Human experts evaluate segments such as this and assign it to one of the sleep stages in {W, N1, N2, N3, R}. In most experiments of this study, U-Time considers only a single EEG channel (for instance C3, as seen above).

Table S.2: U-Time model topology. Layer dimensions below are valid for $i = 3000$, $C = 1$, $T = 35$, $K = 5$. BN = batch normalization. All convolution kernels in layer 1 to 16 (the encoder) are dilated to with 9.

| ID | Layer Type | Output dim | Kernel | Filters | Activation | Pad |
|---|---|---|---|---|---|---|
| 1 | Input | $35 \times 3000 \times 1$ | - | - | - | - |
| 2 | Reshape | $105000 \times 1$ | - | - | - | - |
| 3 | Convolution $\rightarrow$ BN | $105000 \times 16$ | 5 | 16 | ReLU | same |
| 4 | Convolution $\rightarrow$ BN | $105000 \times 16$ | 5 | 16 | ReLU | same |
| 5 | Max Pool | $10500 \times 16$ | 10 | - | - | valid |
| 6 | Convolution $\rightarrow$ BN | $10500 \times 32$ | 5 | 32 | ReLU | same |
| 7 | Convolution $\rightarrow$ BN | $10500 \times 32$ | 5 | 32 | ReLU | same |
| 8 | Max Pool | $1312 \times 32$ | 8 | - | - | valid |
| 9 | Convolution $\rightarrow$ BN | $1312 \times 64$ | 5 | 64 | ReLU | same |
| 10 | Convolution $\rightarrow$ BN | $1312 \times 64$ | 5 | 64 | ReLU | same |
| 11 | Max Pool | $218 \times 64$ | 6 | - | - | valid |
| 12 | Convolution $\rightarrow$ BN | $218 \times 128$ | 5 | 128 | ReLU | same |
| 13 | Convolution $\rightarrow$ BN | $218 \times 128$ | 5 | 128 | ReLU | same |
| 14 | Max Pool | $54 \times 128$ | 4 | - | - | valid |
| 15 | Convolution $\rightarrow$ BN | $54 \times 256$ | 5 | 256 | ReLU | same |
| 16 | Convolution $\rightarrow$ BN | $54 \times 256$ | 5 | 256 | ReLU | same |
| 17 | Up-sample | $216 \times 256$ | 4 | - | - | - |
| 18 | Convolution $\rightarrow$ BN | $216 \times 128$ | 4 | 128 | ReLU | same |
| 19 | Crop & Concat(13, 18) | $216 \times 256$ | - | - | - | - |
| 20 | Convolution $\rightarrow$ BN | $216 \times 128$ | 5 | 128 | ReLU | same |
| 21 | Convolution $\rightarrow$ BN | $216 \times 128$ | 5 | 128 | ReLU | same |
| 22 | Up-sample | $1296 \times 128$ | 6 | - | - | - |
| 23 | Convolution $\rightarrow$ BN | $1296 \times 64$ | 6 | 64 | ReLU | same |
| 24 | Crop & Concat(10, 23) | $1296 \times 128$ | - | - | - | - |
| 25 | Convolution $\rightarrow$ BN | $1296 \times 64$ | 5 | 64 | ReLU | same |
| 26 | Convolution $\rightarrow$ BN | $1296 \times 64$ | 5 | 64 | ReLU | same |
| 27 | Up-sample | $10368 \times 64$ | 8 | - | - | - |
| 28 | Convolution $\rightarrow$ BN | $10368 \times 32$ | 8 | 32 | ReLU | same |
| 29 | Crop & Concat(7, 28) | $10368 \times 64$ | - | - | - | - |
| 30 | Convolution $\rightarrow$ BN | $10368 \times 32$ | 5 | 32 | ReLU | same |
| 31 | Convolution $\rightarrow$ BN | $10368 \times 32$ | 5 | 32 | ReLU | same |
| 32 | Up-sample | $103680 \times 32$ | 10 | - | - | - |
| 33 | Convolution $\rightarrow$ BN | $103680 \times 16$ | 10 | 16 | ReLU | same |
| 34 | Crop & Concat(4, 33) | $103680 \times 32$ | - | - | - | - |
| 36 | Convolution $\rightarrow$ BN | $103680 \times 16$ | 5 | 16 | ReLU | same |
| 35 | Convolution $\rightarrow$ BN | $103680 \times 16$ | 5 | 16 | ReLU | same |
| 36 | Convolution | $103680 \times 5$ | 1 | 5 | TanH | same |
| 37 | Zero padding | $105000 \times 5$ | - | - | - | - |
| 38 | Reshape | $35 \times 3000 \times 5$ | - | - | - | - |
| 38 | Average Pooling | $35 \times 5$ | - | - | - | valid |
| 39 | Convolution | $35 \times 5$ | 1 | 5 | Softmax | same |

**Trainable parameters:** $1,187,589$

Table S.3: Hyperparameters used for all datasets.

| Parameter | Value | Notes |
|---|---|---|
| Optimizer | Adam | We employ a fixed learning rate across |
| *Learning rate -* | $5 \cdot 10^{-6}$ | all datasets. See [Kingma and Ba, |
| $\beta_1$ - | 0.9 | 2014]. |
| $\beta_2$ - | 0.999 | |
| $\epsilon$ - | $1 \cdot 10^{-8}$ | |
| Loss function | Dice loss | See [Sudre et al., 2017, Crum et al., |
| *Regularization -* | None | 2006]. |
| *Class balancing -* | Uniform (None) | |
| Base Topology | 1D U-Net | The input dimensionality is the number. |
| *Input dim -* | 3000 | of data points in a single PSG segment |
| *Window size (T) -* | 35 | (one segment is 30 seconds in typical |
| *Depth -* | 4 | sleep staging, giving input |
| *Up-sampling -* | Nearest neighbour | dimensionality 3000 for sample rate |
| *Activations -* | ReLU | $S = 100$). $T$ is the number of |
| *Conv. kernel size -* | 5 | contiguous segments the model |
| *Conv. kernel dilation size -* | 9 | operates on at once. $T$ may be |
| *Max-pool kernel size -* | $\{10, 8, 6, 4\}$ | dynamically adjusted. Cropping and |
| *Padding -* | True ('same') | zero-padding is needed to decode to |
| *Batch normalization -* | True | dimensions equal to the input, see |
| *Parameters -* | $\approx 1.2 \cdot 10^6$ | [Ronneberger et al., 2015, Odena et al., 2016, Ioffe and Szegedy, 2015, Yu and Koltun, 2015]. |
| Pre-processing | Robust scaling | Record- and channel-wise |
| Post-processing | None | transformation to distribution of |
| Re-sampling (S) | 100 Hz | median 0 and IQR 1. Re-sampling uses polyphase filtering (implementation: `scipy.signal.resample_poly` [Virtanen et al., 2019]). |
| Batch size (B) | 12 | For each member of a batch, a class |
| *Class sampling prob. -* | Uniform | from the label set {W, N1, N2, N3, R} is determined by uniform sampling. A random PSG record that contains the given class is sampled, from which the input window is sampled randomly so that the selected class is present somewhere in the window. |
| Training epochs | $\infty$ | Training continues until 150 |
| *Steps per epoch* | $\lceil L/T/B \rceil$ | consecutive epochs without validation performance improvements. $L$ is the number of 30 second segments in the dataset. |
| Early stopping criteria | Validation F1 | Mean per-class F1 scores (excluding |
| Model selection criteria | Validation F1 | background) computed over all images of a validation epoch. |

Table S.4: U-Time per-record results. Values shown are F1/dice scores computed across all PSG records in each dataset. Each cell displays the mean F1 $\pm$ 1 standard deviation, with the lowest and highest observed F1 score across the records given in the line below indicated by $\downarrow$ and $\uparrow$ respectively.

| Dataset | W | N1 | N2 | N3 | REM |
|---|---|---|---|---|---|
| S-EDF-39 | $0.87 \pm 0.13$ | $0.49 \pm 0.16$ | $0.85 \pm 0.11$ | $0.81 \pm 0.17$ | $0.83 \pm 0.16$ |
| | $\downarrow 0.34 \uparrow 0.99$ | $\downarrow 0.05 \uparrow 0.81$ | $\downarrow 0.25 \uparrow 0.94$ | $\downarrow 0.12 \uparrow 0.96$ | $\downarrow 0.04 \uparrow 0.97$ |
| S-EDF-153 | $0.89 \pm 0.08$ | $0.51 \pm 0.13$ | $0.83 \pm 0.09$ | $0.57 \pm 0.30$ | $0.79 \pm 0.16$ |
| | $\downarrow 0.55 \uparrow 0.99$ | $\downarrow 0.04 \uparrow 0.76$ | $\downarrow 0.40 \uparrow 0.96$ | $\downarrow 0.00 \uparrow 1.00$ | $\downarrow 0.00 \uparrow 0.98$ |
| Physio-18 | $0.78 \pm 0.16$ | $0.57 \pm 0.14$ | $0.81 \pm 0.12$ | $0.69 \pm 0.27$ | $0.78 \pm 0.23$ |
| | $\downarrow 0.00 \uparrow 0.99$ | $\downarrow 0.00 \uparrow 0.87$ | $\downarrow 0.00 \uparrow 0.98$ | $\downarrow 0.00 \uparrow 1.00$ | $\downarrow 0.00 \uparrow 1.00$ |
| DCSM | $0.97 \pm 0.04$ | $0.47 \pm 0.13$ | $0.83 \pm 0.11$ | $0.76 \pm 0.24$ | $0.80 \pm 0.20$ |
| | $\downarrow 0.67 \uparrow 1.00$ | $\downarrow 0.00 \uparrow 0.80$ | $\downarrow 0.29 \uparrow 0.96$ | $\downarrow 0.00 \uparrow 0.97$ | $\downarrow 0.00 \uparrow 0.98$ |
| ISRUC | $0.84 \pm 0.11$ | $0.53 \pm 0.12$ | $0.77 \pm 0.12$ | $0.86 \pm 0.10$ | $0.73 \pm 0.22$ |
| | $\downarrow 0.42 \uparrow 0.97$ | $\downarrow 0.11 \uparrow 0.73$ | $\downarrow 0.07 \uparrow 0.92$ | $\downarrow 0.42 \uparrow 0.99$ | $\downarrow 0.00 \uparrow 0.99$ |
| CAP | $0.70 \pm 0.22$ | $0.28 \pm 0.16$ | $0.74 \pm 0.14$ | $0.79 \pm 0.15$ | $0.73 \pm 0.23$ |
| | $\downarrow 0.00 \uparrow 0.99$ | $\downarrow 0.00 \uparrow 0.66$ | $\downarrow 0.30 \uparrow 0.93$ | $\downarrow 0.10 \uparrow 0.95$ | $\downarrow 0.00 \uparrow 0.95$ |
| SVUH-UCD | $0.73 \pm 0.12$ | $0.46 \pm 0.12$ | $0.75 \pm 0.16$ | $0.79 \pm 0.21$ | $0.67 \pm 0.26$ |
| | $\downarrow 0.52 \uparrow 0.92$ | $\downarrow 0.28 \uparrow 0.66$ | $\downarrow 0.25 \uparrow 0.95$ | $\downarrow 0.00 \uparrow 0.98$ | $\downarrow 0.04 \uparrow 0.93$ |

Table S.5: U-Time ($C = 1$) confusion matrix for dataset Sleep-EDF-39

| | Wake | N1 | N2 | N3 | REM |
|---|---|---|---|---|---|
| Wake | **6980** | 740 | 244 | 22 | 260 |
| N1 | 205 | **1624** | 604 | 15 | 356 |
| N2 | 360 | 615 | **15182** | 982 | 660 |
| N3 | 25 | 7 | 777 | **4892** | 2 |
| REM | 204 | 516 | 523 | 0 | **6474** |

Table S.6: U-Time ($C = 1$) confusion matrix for dataset Sleep-EDF-153

| | Wake | N1 | N2 | N3 | REM |
|---|---|---|---|---|---|
| Wake | **58676** | 5650 | 650 | 40 | 790 |
| N1 | 2364 | **12067** | 5172 | 132 | 1787 |
| N2 | 335 | 5478 | **57437** | 3491 | 2391 |
| N3 | 10 | 69 | 2974 | **9978** | 8 |
| REM | 323 | 2510 | 2280 | 83 | **20639** |

Table S.7: U-Time ($C = 1$) confusion matrix for dataset Physionet-2018

| | Wake | N1 | N2 | N3 | REM |
|---|---|---|---|---|---|
| Wake | **133594** | 20295 | 2473 | 96 | 1487 |
| N1 | 22006 | **83149** | 22744 | 183 | 8896 |
| N2 | 6834 | 32279 | **304191** | 25593 | 8924 |
| N3 | 493 | 214 | 17779 | **84006** | 100 |
| REM | 3165 | 9095 | 6782 | 138 | **97684** |

Table S.8: U-Time ($C = 1$) confusion matrix for dataset DCSM

|      | Wake   | N1     | N2    | N3    | REM   |
|------|--------|--------|-------|-------|-------|
| Wake | **341590** | 5681   | 2326  | 316   | 4396  |
| N1   | 2839   | **11128**  | 4804  | 19    | 2350  |
| N2   | 1888   | 6037   | **94237** | 6586  | 4279  |
| N3   | 195    | 33     | 7156  | **36200** | 53    |
| REM  | 1931   | 1733   | 2522  | 435   | **40205** |

Table S.9: U-Time ($C = 1$) confusion matrix for dataset ISRUC

|      | Wake   | N1    | N2     | N3     | REM   |
|------|--------|-------|--------|--------|-------|
| Wake | **17237**  | 1892  | 512    | 33     | 751   |
| N1   | 1349   | **6505**  | 2316   | 66     | 1254  |
| N2   | 359    | 2649  | **22135**  | 1878   | 1174  |
| N3   | 38     | 10    | 2332   | **14876**  | 26    |
| REM  | 363    | 974   | 938    | 56     | **9589**  |

Table S.10: U-Time ($C = 1$) confusion matrix for dataset CAP

|      | Wake   | N1    | N2     | N3     | REM   |
|------|--------|-------|--------|--------|-------|
| Wake | **14126**  | 1532  | 1779   | 411    | 1004  |
| N1   | 1149   | **1412**  | 997    | 84     | 797   |
| N2   | 1244   | 1351  | **28629**  | 3477   | 2195  |
| N3   | 135    | 32    | 4560   | **19069**  | 296   |
| REM  | 760    | 870   | 2187   | 394    | **13429** |

Table S.11: U-Time ($C = 1$) confusion matrix for dataset SVUH-UCD

|      | Wake   | N1    | N2    | N3    | REM   |
|------|--------|-------|-------|-------|-------|
| Wake | **3537**   | 739   | 227   | 18    | 186   |
| N1   | 783    | **1704**  | 525   | 8     | 383   |
| N2   | 174    | 601   | **5423**  | 410   | 377   |
| N3   | 9      | 7     | 310   | **2328**  | 9     |
| REM  | 207    | 300   | 212   | 22    | **2275**  |

Table S.12: U-Time multi-channel results across 4 datasets. Dataset sizes and evaluation types match those of Table 2 in the main text. Specefic channels used: Sleep-EDF-153: EEG Fpz-Cz, EMG submental, EOG horizontal. Physionet-2018: EEG C3-M2, EEG O1-M2, EMG CHEST. DCSM: EEG C3-M2, EOG E2-M2. ISRUC: EEG C3-M2, EOG ROC-M1.

|           |                     | Global F1 scores |      |      |      |      |      |
|-----------|---------------------|------|------|------|------|------|------|
| Dataset   | Channels            | W    | N1   | N2   | N3   | REM  | mean |
| S-EDF-153 | EEG + EMG + EOG     | 0.92 | 0.51 | 0.82 | 0.72 | 0.84 | 0.76 |
| Physio-18 | 2×EEG + EMG         | 0.83 | 0.58 | 0.83 | 0.79 | 0.83 | 0.77 |
| DCSM      | EEG + EOG           | 0.97 | 0.51 | 0.83 | 0.83 | 0.86 | 0.80 |
| ISRUC     | EEG + EOG           | 0.88 | 0.55 | 0.79 | 0.87 | 0.83 | 0.78 |

Table S.13: Hyperparameter experiments for our re-implemented DeepSleepNet [Kemp et al., 2000] on the Sleep-EDF-39 dataset. The 5-CV hyperparameter experiments were conducted on 25 records only in order to speed up computation. Thus, the performance scores should not be compared directly to the paper re-implementation results (which are based on all 39 records in a 20-CV evaluation), but rather to the *baseline* experiment.

| Experiment | Eval. | Global F1 scores | | | | | |
|---|---|---|---|---|---|---|---|
| | | W | N1 | N2 | N3 | REM | mean |
| Paper re-implementation | 20-CV | 0.86 | 0.41 | 0.87 | 0.83 | 0.81 | 0.76 |
| *Baseline* | 5-CV | 0.85 | 0.39 | 0.86 | 0.89 | 0.79 | 0.76 |
| Smaller CNN filters | 5-CV | 0.84 | 0.31 | 0.86 | 0.87 | 0.77 | 0.73 |
| Larger CNN filters | 5-CV | 0.84 | 0.30 | 0.87 | 0.87 | 0.76 | 0.73 |
| Two CNN layers | 5-CV | 0.84 | 0.26 | 0.85 | 0.85 | 0.76 | 0.71 |
| Four CNN layers | 5-CV | 0.84 | 0.31 | 0.86 | 0.89 | 0.78 | 0.74 |
| One RNN layer | 5-CV | 0.85 | 0.35 | 0.86 | 0.88 | 0.78 | 0.74 |
| Three RNN layers | 5-CV | 0.76 | 0.41 | 0.85 | 0.85 | 0.75 | 0.72 |
| Short sequences ($T = 10$) | 5-CV | 0.83 | 0.31 | 0.86 | 0.87 | 0.75 | 0.72 |
| Long sequences ($T = 50$) | 5-CV | 0.83 | 0.34 | 0.86 | 0.86 | 0.74 | 0.73 |
| LSTM $\rightarrow$ GRU | 5-CV | 0.84 | 0.35 | 0.86 | 0.87 | 0.74 | 0.73 |
| LSTM 64 cells | 5-CV | 0.85 | 0.32 | 0.85 | 0.84 | 0.78 | 0.73 |
| LSTM 256 cells | 5-CV | 0.85 | 0.31 | 0.86 | 0.86 | 0.77 | 0.73 |
| Dropout $\rightarrow$ Zoneout (5%) | 5-CV | 0.80 | 0.34 | 0.85 | 0.88 | 0.77 | 0.73 |
| Dropout $\rightarrow$ Zoneout (10%) | 5-CV | 0.80 | 0.39 | 0.84 | 0.87 | 0.77 | 0.73 |
| CNN filter size 3-ensemble | 5-CV | 0.86 | 0.34 | 0.87 | 0.88 | 0.79 | 0.75 |
| FPZ+CZ+EOG ensemble | 5-CV | 0.91 | 0.40 | 0.89 | 0.85 | 0.87 | 0.78 |

Table S.14: Hyperparameter experiments for our re-implemented DeepSleepNet [Kemp et al., 2000] on the DCSM dataset. The hyperparameter experiments were conducted on a 100-records subset of the DCSM dataset to speed up computation. Thus, performance scores should not be compared to the results in Table 2 directly, but rather to the *baseline* experiment.

| Experiment | Eval. | Global F1 scores | | | | | |
|---|---|---|---|---|---|---|---|
| | | W | N1 | N2 | N3 | REM | mean |
| *Baseline* | 5-CV | 0.95 | 0.33 | 0.81 | 0.77 | 0.80 | 0.73 |
| Smaller CNN filters | 5-CV | 0.95 | 0.37 | 0.80 | 0.79 | 0.81 | 0.74 |
| Larger CNN filters | 5-CV | 0.94 | 0.34 | 0.81 | 0.77 | 0.80 | 0.73 |
| LSTM $\rightarrow$ GRU | 5-CV | 0.95 | 0.33 | 0.80 | 0.76 | 0.80 | 0.73 |
| Short sequences ($T = 10$) | 5-CV | 0.95 | 0.32 | 0.80 | 0.75 | 0.78 | 0.72 |
| Long sequences ($T = 50$) | 5-CV | 0.95 | 0.32 | 0.79 | 0.78 | 0.79 | 0.73 |
| Four input signals ensemble | 5-CV | 0.96 | 0.36 | 0.83 | 0.80 | 0.81 | 0.75 |

Figure S.2: Expanded structural overview of the U-Time architecture.