[Reviews · NeurIPS 2019]

Reviewer 1



-- The results presented in Table 1 are a bit mixed. While U-Time does not do worse than any other algorithm to which it had been directly compared, many methods were not implemented for many of the datasets. I believe it remains to be demonstrated that a significant advance to automated sleep staging has been made. -- The architecture of U-Time is a bit hard to understand from the diagrams and descriptions given. -- Would the time-series network architecture possibly generalize to segmenting other types of time-series data for which some event segments are very rare?

Reviewer 2



- Originality: The authors tackled the well-known sleep staging segmentation task with a new method U-Time for time-series segmentation, which derived from the commonly-used image segmentation model U-Net. Also, different from the previous studies mainly focus on RNN-based architecture for time-series analysis, the authors proposed a 1D convolution and FCN-based model which has less limitation on temporal resolution and distributed computation. The sleep staging literature is well-cited but needs more comparison (better in the related work section). The methods used for comparison in Table 1 are better to be discussed in the beginning. The references of time-series segmentation/classification should also be added from the methodology perspective. - Quality: This is a complete study, and the proposed U-Time method is technically sound for the time-series segmentation problem. No theoretical analysis is provided but the empirical results demonstrate that the method is promising and support the claims that it works better than other recurrent-based methods in some conditions. The authors also performed detailed hyperparameter experiments in the supplementary material. The weakness of the study is that the baseline for each dataset is not unified---in S-EDF-39 there are many methods for comparison but for others, there are one (but different across datasets) or no baseline to compare with. - Clarity: The paper is well-written and well-organized. The dataset description may be organized into a table instead of using a full page to describe them. Some qualitative analysis would be helpful for model interpretability. - Significance: The results of using U-Time on multiple sleep staging datasets demonstrate that the proposed framework is useful for time-series segmentation in sleep staging problem. This may be potentially helpful for other, general time-series segmentation tasks as well. From a clinical perspective, good PSG/EEG segmentation and classification can assist neurologists to diagnose sleep-related diseases in a shorter time, which may optimize the clinical workflow.

Reviewer 3



Post response: The authors did a great job responding to my question about architecture decisions and I recommend summarizing some of this in the paper. On the other hand, I think the authors may have missed my point about error analysis. When I suggested using confusion matrices, I was not necessarily suggesting including them in the paper, I was suggesting using as a tool to help explain the apparent systematic differences in predictive behavior between the models. That is, the authors have convinced me that their model performs better on N1 segments, but I have no idea why that is the case (which I consider the more interesting scientific question). I think the paper should be accepted as is, but could be improved by including this type of error analysis. Also, Figure 1 in the author response is a great figure that I recommend including in the main paper. -------------------------------------------- Overall: In this work, the authors propose a convolutional neural net based model for sleep staging and test this model on several sleep staging datasets. Overall, I found this to be a well-executed application paper with a meaningful, well-explained application, a reasonable approach, and a convincing evaluation. My main comments are that the model seems rather arbitrary and the evaluations could use some error analysis. Major comments: 1. The model, while reasonable, seemed very arbitrary. If the architecture was based on some theory or intuition for what would work in this application, the authors should explain this theory. If the authors tried several models with less success, then sensitivity to these choices should be described. For example, in the Decoder section, the authors specify four transposed-convolution blocks with specific kernel sizes. How were these values selected? If, as I suspect, there was some architecture searching done, details of this search (e.g. sensitivity to the number of transposed-convolution blocks) should be included in the supplementary material. 2. I think the results section would heavily benefit from some error analysis. As it stands, the authors identify certain systematic differences in the errors between models, but never explain why those occur. Perhaps confusion matrices (some of which the authors included in the supplementary materials) would be useful for this. As the authors note, the most glaring such systematic difference appears to be that U-time performs better on N1 segments, which are also the hardest to recognize. Why is this the case? Minor comments: 1. Lines 43-45 "U-Time as opposed to...": What is the intuition behind this dataset invariance? 2. Figure 1 is hard to see in black and white. I recommend using a color palette that is good for color blindness (e.g. https://davidmathlogic.com/colorblind/) 3. Could you expand on the "confidence scores" in lines 118-121? In particular, how are they generated and do they have any specific interpretation beyond general "confidence" (e.g. as probabilities)? 4. Lines 177-179: Could you expand on the hyperparameter selection procedure? Why not re-tune for each dataset? Was cross-validation used for tuning on S-EDF-39?

[Author Response · NeurIPS 2019]

**General:** Thank you for your feedback. The main motivation/intuition behind our approach is the excellent performance we achieved with U-Nets for image segmentation. The theoretical/conceptual contribution is to exploit the similarity between sleep staging – and time series segmentation in general – and image segmentation. Still, U-Time is not simply a 1D U-Net (e.g., because of the *segment classifier*, dilated convolutions, normalization, NN-upsampling, . . . ).

We did not run our DeepSleepNet implementation as a baseline on all datasets, because of lacking compute resources and, given the extensive experiments reported in the supplement, because we were (and still are) sure that U-Time would clearly outperform it. We now understand the value of explicitly stating the additional baseline results and are currently running the corresponding experiments. We would like to stress that we are comparing a system developed on a single design dataset (S-EDF-39) across several datasets. In contrast, the baselines come from papers that propose one system for one dataset. Thus, the cited results are prone to (unintentional) method overfitting. This is why we regard our results as a significant advance to automated sleep staging resaerch and clinical practice.

The N1 class is hard because it is rare and difficult to clearly separate from other classes, even for human experts as can be seen in Table 1. N1 lies between wake and deeper non-REM sleep, and the transitions are gradual.

**Rev. 1:** Yes, we used only one channel for a fair comparison to existing literature. We are currently evaluating the use of multiple channels and would be happy to report results in the supplement. We will include a supplementary figure showing an examplary confidence score (softmax) output for a $C = 3$ multi-channel input, see Figure 1 below and our response to Rev. 2.

Details of the layers can be found in the supplement Table S.2. We will add another, larger figure illustrating the U-shape and the three parts of the architecture. We will make the code available for maximum reproducibility. The classes in our application are already very imbalanced, so U-Time may also work for very rare events.

**Rev. 2:** We are happy to restructure the manuscript according to your suggestions. Regarding confidence scores, qualitative analysis, and interpretability: It is particularly interesting to inspect the outputs of U-Time when the (freely adjustable) segmentation frequency is set to match to the input signal frequency, see Figure 1 below. The sleep stage scores indeed show human interpretable patterns even on short timescales. We believe that this special property of U-Time will allow for a better analysis of sleep stage transitions in healthy and diseased populations. We will discuss this in the main text including cases where the model fails to predict the true sleep stages and add the figure below to the supplementary material.

AASM stands for *American Academy of Sleep Medicine* (see line 129, sorry, we forgot to add the abbreviation). $T$ is the number of fixed-length connected segments (each typically 30s) input to the model (line 68), which we will recall in line 169. $B$ is the batch size, which we will introduce properly.

**Rev. 3:** Regarding architecture choice: The U-Net part of our architecture is based on an architecture that worked extremely well in medical image segmentation across a wide range of problems. There is a paper at the upcoming MICCAI showing this, which we cannot cite without revealing authors of the current submission (we can share a preprint with the area chair). In our submission (lines 91–97), we discuss the sizes of the kernels in relation to the time windows the layers see – an important design criterion. We selected them based on our physiological understanding of sleep staging. We regard it as a big advantage that we did not extensively tune our architecture to the tasks. The fact that this was not necessary demonstrates the soundness of the basic approach and the robustness of the implementation. It is important for us that our results are not artefacts resulting from (unintentional) overfitting through architecture / hyperparameter tuning. In contrast, as shown in the supplement, we un-

Figure 1: U-Time confidence scores (softmax output) over $T = 11$ segments (30s each) for $C = 3$ input channels (EEG, EMG and EOG). The freely adjustable segmentation frequency is set to match the input signal frequency.

successfully tried to tune competing methods to reach the performance of U-Time. Tuning the U-Time architecture and hyperparameters may improve the results. However, we assume that adding different channels or other input modalities is more important.

Confusion matrices (CMs) for U-Time on all datasets are given in the supplement, and we are happy to add CMs for other methods. We will improve the colour palette as suggested. See the new figure above for an illustration of the confidence scores (which we will explain better in the main text). We used 5-fold cross-validation on the design dataset to choose between two loss functions (cross entropy and the dice loss).

[Meta-Review · NeurIPS 2019]

The authors have presented a clear rebuttal and the reviewers have updated their reviews based on the rebuttal. There were still some aspects which the reviewers thought still needed to be discussed and shown in the paper. In discussion with the reviewers, we thought that given the empirical nature of the experiment design, interpretability and error analysis (as suggested by R3) are important. Less model architecture sensitivity analysis was compensated for by the authors showing that the proposed method can be applied to multiple tasks with reasonable performance. However, the magnitude of improvement wasn't obvious as the baseline comparison is not clear and completed.